# Deficiency of parkin and PINK1 impairs age-dependent mitophagy in *Drosophila*

**Tom Cornelissen[1], Sven Vilain[2,3], Katlijn Vints[4], Natalia Gounko[4], Patrik Verstreken[2,3], Wim Vandenberghe[1,5]\***

[1]Laboratory for Parkinson Research, Department of Neurosciences, Leuven, Belgium; [2]Department of Neurosciences, KU Leuven, Leuven, Belgium; [3]VIB-KU Leuven Center for Brain & Disease Research, Leuven, Belgium; [4]VIB Electron Microscopy Platform and BioImaging Core, Department of Neurosciences, VIB-KU Leuven Center for Brain and Disease Research, KU Leuven, Leuven, Belgium; [5]Department of Neurology, University Hospitals Leuven, Leuven, Belgium

**Abstract** Mutations in the genes for PINK1 and parkin cause Parkinson's disease. PINK1 and parkin cooperate in the selective autophagic degradation of damaged mitochondria (mitophagy) in cultured cells. However, evidence for their role in mitophagy in vivo is still scarce. Here, we generated a *Drosophila* model expressing the mitophagy probe mt-Keima. Using live mt-Keima imaging and correlative light and electron microscopy (CLEM), we show that mitophagy occurs in muscle cells and dopaminergic neurons in vivo, even in the absence of exogenous mitochondrial toxins. Mitophagy increases with aging, and this age-dependent rise is abrogated by PINK1 or parkin deficiency. Knockdown of the *Drosophila* homologues of the deubiquitinases USP15 and, to a lesser extent, USP30, rescues mitophagy in the parkin-deficient flies. These data demonstrate a crucial role for parkin and PINK1 in age-dependent mitophagy in *Drosophila* in vivo.
DOI: https://doi.org/10.7554/eLife.35878.001

**\*For correspondence:**
wim.vandenberghe@uzleuven.be

## Introduction

Loss-of-function mutations in *PARK2* and *PINK1*, which encode the cytosolic E3 ubiquitin ligase parkin and the mitochondrial ubiquitin kinase PINK1, respectively, are the most prevalent recessive causes of Parkinson's disease (PD), an age-dependent neurodegenerative disorder (*Corti et al., 2011*). Parkin and PINK1 cooperate in the selective autophagic degradation of damaged mitochondria (mitophagy) (*Pickrell and Youle, 2015*). Upon mitochondrial damage, PINK1 is stabilized on the outer mitochondrial membrane (OMM) and phosphorylates ubiquitin and parkin at their respective S65 residues. This activates parkin to catalyze ubiquitination of many OMM proteins that are then either degraded by the proteasome or act as signals for autophagic clearance of the mitochondrion (*Pickrell and Youle, 2015*). Parkin-mediated mitochondrial ubiquitination and mitophagy are counteracted by specific deubiquitinases (*Bingol et al., 2014*; *Cornelissen et al., 2014*). PINK1/parkin-mediated mitophagy can be triggered in a variety of neuronal and non-neuronal cells in vitro, typically by exposure to mitochondrial depolarizing agents (*Narendra et al., 2008*; *Geisler et al., 2010*; *Cai et al., 2012*; *Ashrafi et al., 2014*; *Cornelissen et al., 2014*; *Oh et al., 2017*). Along with other mitochondrial quality control mechanisms PINK1/parkin-mediated mitophagy contributes to the maintenance of a healthy mitochondrial network (*Pickles et al., 2018*).

Despite the wealth of mechanistic information on PINK1/parkin-mediated mitophagy in cultured cells, questions still surround the existence and physiological relevance of this pathway in vivo (*Cummins and Götz, 2018*; *Whitworth and Pallanck, 2017*). Direct evidence for the occurrence of PINK1/parkin-mediated mitophagy in vivo is still scarce. Ubiquitin phosphorylated at S65 (pS65-Ub), a biomarker of PINK1 activity, accumulates in brains from elderly human subjects (*Fiesel et al.,*

**eLife digest** Parkinson's disease is a brain disorder where certain nerve cells slowly die, and the symptoms gradually worsen over time. While the risk of developing the condition increases with age, in certain patients the illness is caused by defects in two proteins, PINK1 and parkin.

PINK1 and parkin help to manage mitochondria, the compartments in our cells that create molecules that serve as the energy currency for nearly all biological processes. When mitochondria get damaged, they release harmful substances that can kill their host cell. To prevent this, PINK1 and parkin can start a process known as mitophagy, which allows the cell to safely dispose of these dangerous mitochondria.

Yet, mitophagy that is triggered by PINK1 and parkin has only been observed in cells grown in the laboratory; there is very little direct evidence that it also takes place in living organisms. If this mechanism does not happen in animals, then it is probably not relevant to Parkinson's disease.

Here, Cornelissen et al. genetically engineered fruit flies that carry a fluorescent marker which helps to track when and where damaged mitochondria are destroyed by a cell. The experiments revealed that mitophagy took place in muscles and in brain tissues. As the animals grew older, mitophagy became more frequent. However, this increase in mitophagy was not seen in insects that did not have PINK1 and parkin. These results showed that the role of PINK1 and parkin in mitophagy is not restricted to cells grown artificially.

The fruit flies designed by Cornelissen et al. will be useful to investigate how PINK1 and parkin keep cells healthy by disposing of harmful mitochondria in living organisms. Ultimately, this may help to develop treatments that slow down the development of Parkinson's disease.
DOI: https://doi.org/10.7554/eLife.35878.002

*2015*) and from mice with a genetic defect in mitochondrial DNA proofreading (*Pickrell et al., 2015*), suggesting that PINK1-mediated mitophagy is induced by aging and accumulation of mitochondrial damage. However, a recent study reported that mitophagy occurs independently of PINK1 in the mouse (*McWilliams et al., 2018*). In *Drosophila*, deficiency of PINK1 or parkin causes reduced life span and severe flight muscle degeneration with accumulation of swollen mitochondria (*Greene et al., 2003*; *Pesah et al., 2004*; *Clark et al., 2006*; *Park et al., 2006*). In parkin mutant flies, half-lives of mitochondrial proteins are drastically prolonged, consistent with a role for parkin in mitophagy, but the effect of PINK1 on mitochondrial protein turnover is much more limited (*Vincow et al., 2013*). Overall, it remains to be directly demonstrated whether PINK1 and parkin can target damaged mitochondria to lysosomes in vivo.

Recently, the mt-Keima reporter was developed to quantitatively image mitophagy in vitro and in vivo (*Katayama et al., 2011*; *Sun et al., 2015*). Here, we used live mt-Keima imaging and correlative light and electron microscopy (CLEM) to assess mitophagy in *Drosophila* in vivo and to determine the role of parkin and PINK1 in this pathway.

## Results

### Mitophagy occurs in *Drosophila* flight muscle and increases with aging

The phenotype of parkin- and PINK1-deficient flies is especially striking in flight muscle (*Greene et al., 2003*; *Pesah et al., 2004*; *Clark et al., 2006*; *Park et al., 2006*). We therefore focused on the role of mitophagy in this tissue. We generated flies that express mt-Keima specifically in muscle (*mef-2-Gal4*). The mt-Keima reporter is a mitochondrially targeted form of Keima, a fluorescent protein that resists degradation by lysosomal proteases (*Katayama et al., 2011*). The peak of the mt-Keima excitation spectrum shifts when mitochondria are delivered to acidic lysosomes, which allows dual-excitation ratiometric quantification of mitophagy (*Katayama et al., 2011*) (*Figure 1A*). Transmission electron microscopy (TEM) analysis showed that mt-Keima expression did not change the morphology of muscle mitochondria (*Figure 1B*). The mt-Keima signal extensively colocalized with the mitochondrial protein ATP synthase β, confirming that mt-Keima was properly targeted to mitochondria (*Figure 1C*). Interestingly, a small subset of mt-Keima structures had high 543 nm/458 nm ratio values, indicative of an acidic environment (*Figure 1D*). These high 543/458

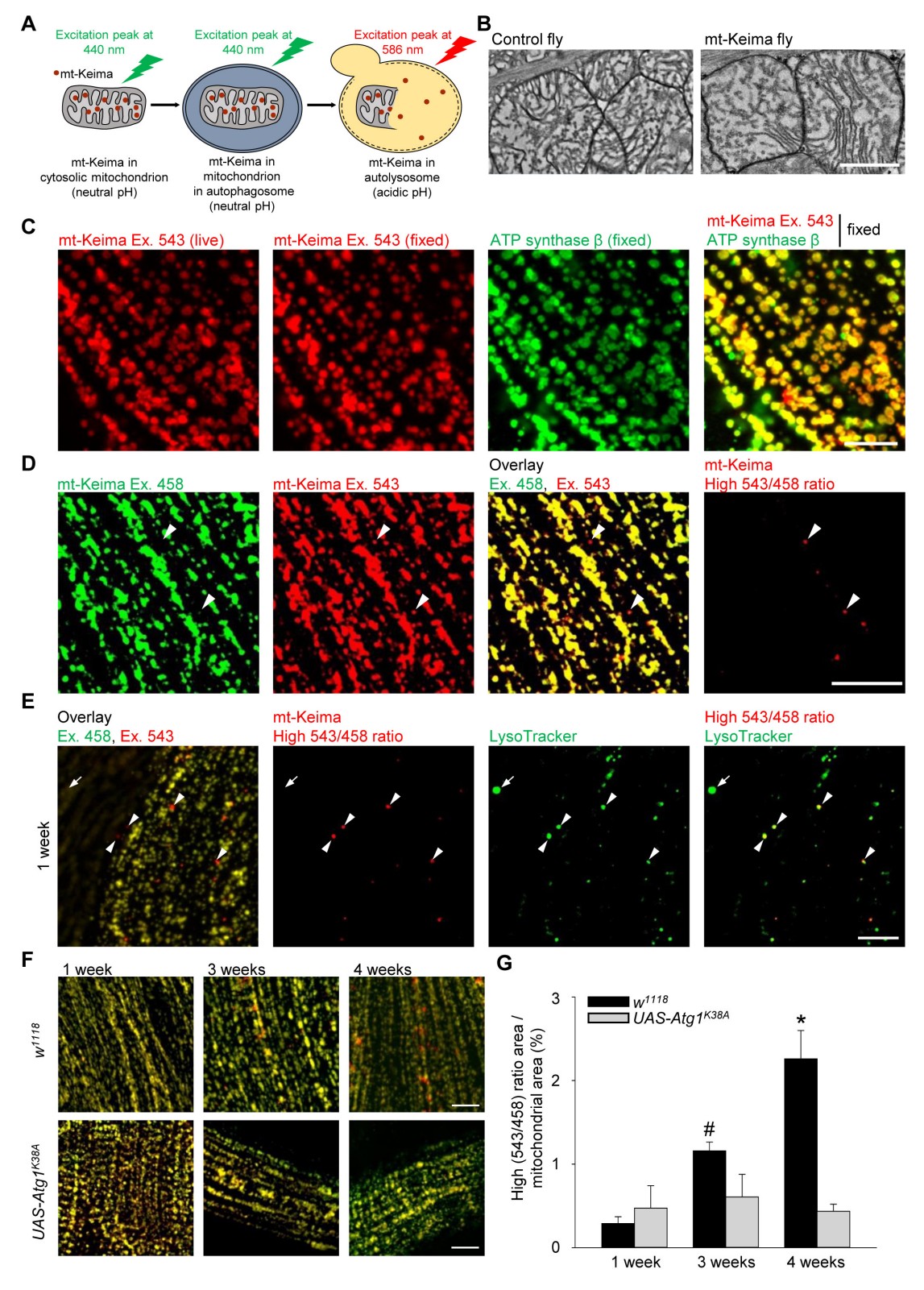

**Figure 1.** Mitophagy in *Drosophila* flight muscle increases with aging. (**A**) Schematic of mt-Keima imaging. The fluorescent mt-Keima protein is targeted to the mitochondrial matrix and exhibits pH-dependent excitation. The excitation peak of mt-Keima shifts from 440 nm to 586 nm when mitochondria are delivered to acidic lysosomes, where it resists degradation by lysosomal proteases. (**B**) Transmission electron micrographic images of mitochondria from control (*w^1118^;;*) and mt-Keima-expressing indirect flight muscle cells under control of the *mef-2-GAL4* driver (*w^1118^;; UAS-mt-Keima,*

*Figure 1 continued*

*mef-2-GAL4/+*). Scale bar, 1 µm. (C) Confocal images of 1-week-old indirect flight muscle expressing mt-Keima under control of the *mef-2-GAL4* driver (*w[1118];; UAS-mt-Keima, mef-2-GAL4/+*). Images show mt-Keima emission at 543 nm excitation (Ex.) before and after fixation, immunostaining for ATP synthase β after fixation, and overlay. Scale bar, 10 µm. (D) Images of 1-week-old indirect flight muscle expressing mt-Keima under control of the *mef-2-GAL4* driver (*w[1118];; UAS-mt-Keima, mef-2-GAL4/+*). Images show live mt-Keima emission at 458 and 543 nm Ex., overlay of mt-Keima emissions at 458 and 543 nm Ex., and puncta with high 543 nm/458 nm ratio values. *Arrowheads* indicate examples of 'acidic' mt-Keima puncta. Scale bar, 10 µm. (E) Confocal images of mt-Keima-expressing indirect flight muscle labeled with LysoTracker (100 nM), showing colocalization of 'acidic' mt-Keima puncta with lysosomes (*arrowheads*). Arrow indicates an example of a lysosome devoid of mt-Keima signal. Scale bar, 10 µm. (F) Overlay of mt-Keima emission at Ex. 458 (green) and 543 nm (red) in *w[1118]* or Atg1[K38A]-overexpressing (*w[1118];; UAS-mt-Keima, mef-2-GAL4/UAS-Atg1[K38A]*) indirect flight muscle of 1-, 3- and 4-week-old flies. Scale bar, 10 µm. (G) High (543/458) ratio area/total mitochondrial area was quantified as an index of mitophagy (n = 4–6 flies per condition). In each fly, 7 random 2500 µm$^2$ fields were analyzed. *p=0.02 compared with all other conditions; # p=0.02 compared with 1-week-old *w[1118]* flies.
DOI: https://doi.org/10.7554/eLife.35878.003

The following source data is available for figure 1:

**Source data 1.** Quantification of mitophagy (high [543/458] ratio area/total mitochondrial area) in indirect flight muscle of *w[1118]* or Atg1[K38A]-overexpressing flies at 1, 3 and 4 weeks.
DOI: https://doi.org/10.7554/eLife.35878.004

ratio mt-Keima puncta colocalized with the lysosomal dye LysoTracker (*Figure 1E*). Thus, the high 543/458 ratio mt-Keima puncta probably represented mitochondria that had been delivered to lysosomes. The abundance of these puncta was very low in muscle cells of 1-week-old adult flies, but strongly rose with aging, showing an approximately tenfold increase by the age of 4 weeks (*Figure 1F,G*). To assess the role of autophagy in the biogenesis of the high 543/458 ratio mt-Keima puncta, we overexpressed a kinase-dead version of Atg1 (Atg1[K38A]), the homologue of mammalian ULK1 (*Toda et al., 2008*). Atg1/ULK1 is needed in the early steps of autophagosome formation and is also involved in mitophagy (*Itakura and Mizushima, 2010*; *Itakura et al., 2012*). When overexpressed, kinase-dead Atg1 exerts dominant-negative effects (*Scott et al., 2007*). Overexpression of Atg1[K38A] suppressed the high levels of mitophagy observed in aged muscle cells (*Figure 1F,G*). The low residual level of high 543/458 ratio mt-Keima puncta that persisted after Atg1[K38A] overexpression, may result from Atg1-independent mechanisms of autophagy induction (*Braden and Neufeld, 2016*).

## CLEM reveals the ultrastructure of 'acidic' mt-Keima puncta

To demonstrate the occurrence of mitophagy in *Drosophila* indirect flight muscle more definitively, we resorted to CLEM (*Bishop et al., 2011*). We first performed live mt-Keima imaging of muscle cells to identify regions of interest that contained both 'acidic' and 'neutral pH' mt-Keima structures (*Figure 2A*). After fixation, we burned laser marks around the regions of interest using near-infrared branding (NIRB) to be able to re-identify these regions after processing for TEM (*Figure 2B–D*). Interestingly, 'neutral pH' mt-Keima structures colocalized with typical mitochondria at the TEM level, while 'acidic' mt-Keima puncta showed remarkable overlay with smaller organelles with the characteristic electron-dense appearance of lysosomes (*Figure 2E–N*). Many of the mt-Keima-positive lysosomes contained densely packed concentric membranes surrounding a dense core, giving them the appearance of multilamellar bodies (*Figure 2N*). Multilamellar bodies are lysosomal organelles that are found in various cell types in physiological conditions but also accumulate in lysosomal storage diseases (*Blanchette-Mackie, 2000*; *Hariri et al., 2000*; *Lajoie et al., 2005*).

## Loss of Parkin and PINK1 impairs mitophagy in *Drosophila* flight muscle and dopaminergic neurons

We then crossed the mt-Keima-expressing flies with *PINK1* loss-of-function mutant flies (*PINK1[B9]*) and *parkin* RNAi flies (*Park et al., 2006*; *Cornelissen et al., 2014*). The low level of mitophagy in 1-week-old adult flies was not affected by deficiency of PINK1 or parkin (*Figure 3A,B*). However, the age-dependent increase in mitophagy in 3- and 4-week-old control flies was completely abolished in the *PINK1[B9]* and *parkin* RNAi flies (*Figure 3A,B*).

The DUBs USP15 and USP30 oppose parkin-mediated mitochondrial ubiquitination and mitophagy in cultured human cells. Interestingly, the yeast homologue of USP15 (Ubp12) deubiquitinates

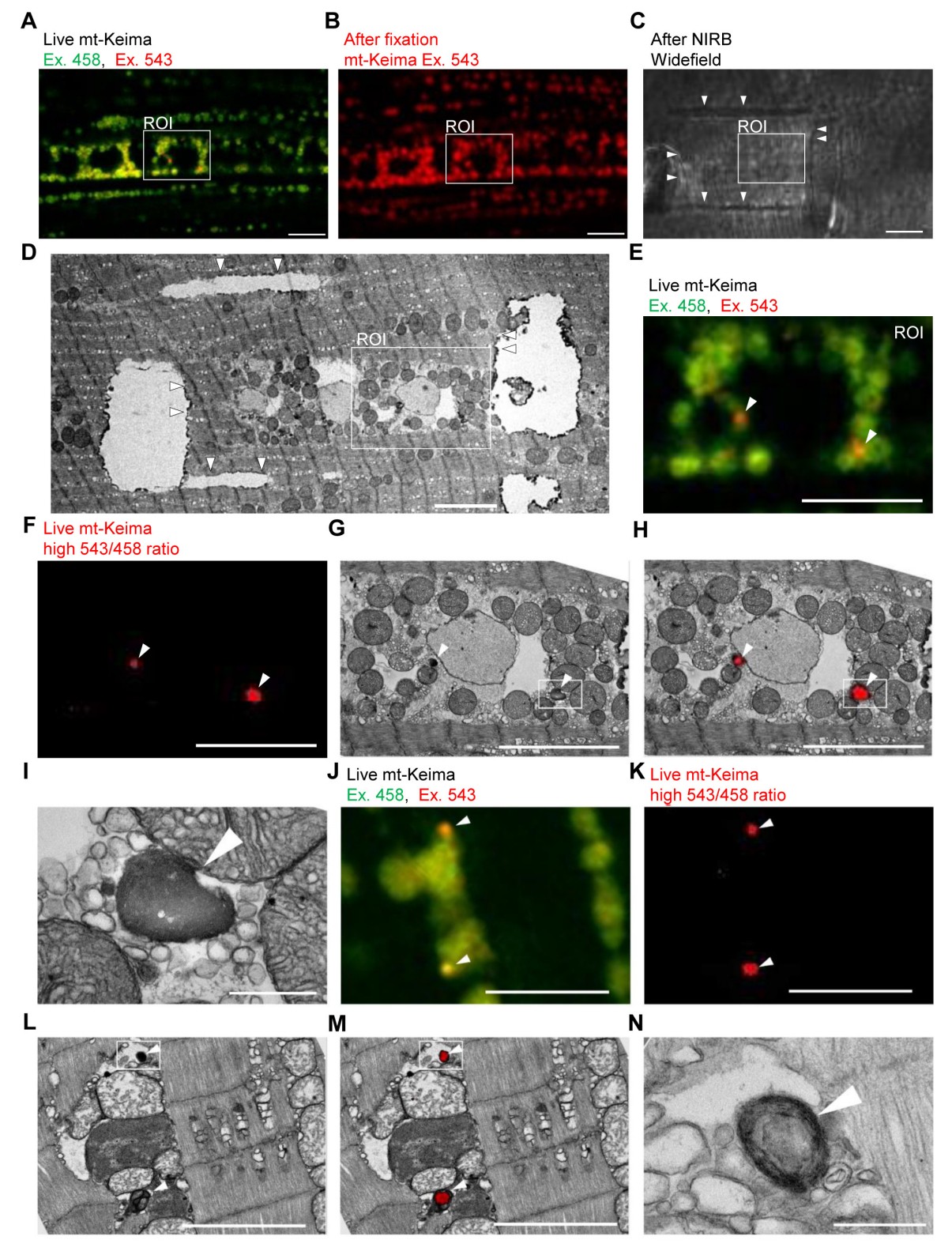

**Figure 2.** Correlative light and electron microscopy (CLEM) reveals the ultrastructure of 'acidic' mt-Keima puncta. (A–D) Overview of the CLEM procedure from live mt-Keima imaging to EM. (A) Overlay of live mt-Keima emission at 458 (green) and 543 nm (red) excitation (Ex.) in indirect flight muscle tissue of a 2-week-old control fly (*w[1118];; UAS-mt-Keima, mef-2-GAL4/+*). Boxed area shows the region of interest (ROI) containing two high 543/458 ratio ('acidic') mt-Keima dots. (B) After fixation the same ROI as in (A) is re-identified based on muscle fiber morphology (mt-Keima emission at Ex.
*Figure 2 continued on next page*

*Figure 2 continued*

543 nm). (C) Widefield image after near-infrared branding (NIRB) around the ROI (laser marks indicated by *arrowheads*). (D) EM image of the same ROI as in (A–C), surrounded by NIRB marks (indicated by *arrowheads*). (E–H) Magnification of the same ROI shown in (A–D). *Arrowheads* in (E–H) indicate 'acidic' mt-Keima dots that colocalize with lysosomes on EM. (E) Overlay of live mt-Keima emission at 458 (green) and 543 nm (red) Ex. (F) High 543/458 ratio dots. (G) EM. (H) Overlay of EM and high 543/458 ratio mt-Keima image. (I) Further magnification of the boxed region in (G–H). (J–N) Additional CLEM example in muscle tissue of a 1-week-old control fly ($w^{1118}$;; UAS-mt-Keima, mef-2-GAL4/+). *Arrowheads* in (J–M) indicate 'acidic' mt-Keima dots that colocalize with lysosomes on EM. (J) Live mt-Keima emission at 458 (green) and 543 nm (red) Ex. in muscle. (K) High 543/458 ratio dots. (L) EM. (M) Overlay of EM and high 543/458 ratio mt-Keima image. (N) Further magnification of the boxed region in (L–M). Scale bars in (I) and (N), 500 nm. All other scale bars, 5 μm.

DOI: https://doi.org/10.7554/eLife.35878.005

the MOM protein Fzo1, the yeast homologue of the mammalian parkin substrates mitofusin 1 and 2, pointing to a conserved role for USP15 in deubiquitination at the MOM (*Anton et al., 2013*; *SimoesSimões et al., 2018*). Knockdown of the *Drosophila* homologs of *USP15* (*CG8334*, hereafter called *dUSP15*) and *USP30* (*CG3016*, hereafter *dUSP30*) largely rescues the mitochondrial defects of parkin-deficient fly muscle in vivo (*Bingol et al., 2014*; *Cornelissen et al., 2014*). To assess the effects of dUSP15 and dUSP30 on PINK1/parkin-mediated mitophagy in vivo, we knocked down dUSP15 and dUSP30 in mt-Keima-expressing flies using RNAi. Levels of *dUSP15* mRNA in *dUSP15* RNAi lines were 57.6 ± 2.5% of control levels (*n* = 6), and *dUSP30* mRNA levels in *dUSP30* RNAi lines were 16.4 ± 5.8% of controls (*n* = 6). Knockdown of *dUSP15* and, to a lesser extent, *dUSP30* rescued the mitophagy defect of two different *parkin* RNAi fly lines (*Figure 3C,D*). The mitophagy defect of *PINK1^{B9}* flies was partially rescued by *dUSP15* knockdown, but unaffected by *dUSP30* knockdown (*Figure 3C,D*).

Dopaminergic neurons of *parkin* mutant *Drosophila* also accumulate abnormal mitochondria (*Burman et al., 2012*). We therefore expressed mt-Keima in dopaminergic neurons (*TH-GAL4*). As in muscle cells, mitophagy increased with aging (*Figure 4A,B*). Loss of parkin suppressed mitophagy in 4-week-old flies (*Figure 4A,B*), and this deficit was rescued by *dUSP15* knockdown (*Figure 4C,D*).

## Discussion

Our data show that mitophagy occurs in *Drosophila* flight muscle and dopaminergic neurons in vivo, even in the absence of exogenous mitochondrial toxins. Mitophagy in these cells rises with aging, and this age-dependent increase is abrogated by PINK1 or parkin deficiency. Thus, PINK1 and parkin appear to be particularly important for mitophagy during aging, which may explain why loss of these proteins in patients causes an age-dependent neurodegenerative disease rather than a congenital or developmental disorder. Consistent with this, pS65-Ub, a readout of PINK1 activity, is almost undetectable in postmortem brains from neurologically normal, young human subjects, but accumulates in brains from elderly individuals (*Fiesel et al., 2015*).

A recent study reported that PINK1 is dispensable for mitophagy in the mouse in vivo (*McWilliams et al., 2018*). This conclusion was based on imaging using the *mito*-QC reporter, which differs from the mt-Keima probe used in our study. *mito*-QC is targeted to the OMM, whereas mt-Keima is localized to the mitochondrial matrix. Upon mitochondrial damage, activated parkin ubiquitinates a wide variety of OMM proteins, many of which are then extracted from the OMM and degraded by the proteasome before engulfment of the mitochondrion into an autophagosome (*Tanaka et al., 2010*; *Chan et al., 2011*; *Sarraf et al., 2013*). The *mito*-QC probe consists of a tandem mCherry-GFP tag fused to a large portion of the OMM protein FIS1 (*McWilliams et al., 2016*), which is itself a substrate of parkin (*Chan et al., 2011*; *Sarraf et al., 2013*). If *mito*-QC is ubiquitinated by parkin and extracted from the OMM, this would prevent *mito*-QC delivery to lysosomes and reduce its sensitivity as a reporter for PINK1/parkin-mediated mitophagy. An alternative explanation for the discrepant results could be that mice may not accumulate sufficient mitochondrial damage in their lifespan to activate PINK1/parkin-mediated mitophagy. The abundance of pS65-Ub in cerebral cortex is very low in wild-type mice, but is substantially higher in mice expressing a proofreading-deficient version of mitochondrial DNA polymerase γ (*Pickrell et al., 2015*). This suggests that a 'second hit' in addition to aging may be required to induce PINK1/parkin-mediated

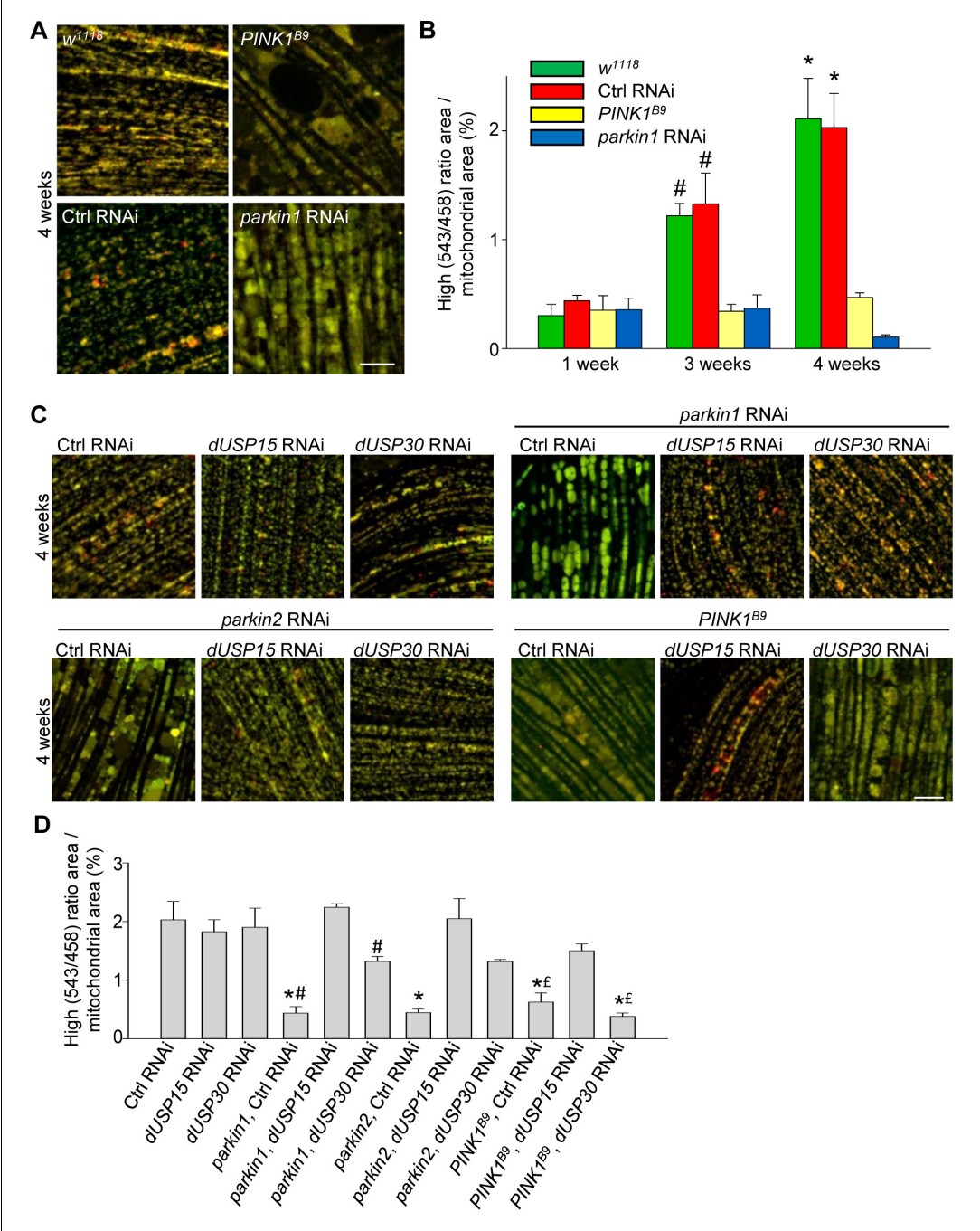

**Figure 3.** Deficiency of parkin and PINK1 impairs mitophagy in *Drosophila* flight muscle. (**A**) Overlay of live mt-Keima emission at 458 (green) and 543 nm (red) excitation in indirect flight muscle from 4-week-old $w^{1118}$ ($w^{1118}$;; UAS-mt-Keima, mef-2-GAL4/+), control (Ctrl) RNAi ($w^{1118}$; UAS-Ctrl RNAi/+; UAS-mt-Keima, mef-2-GAL4/+), PINK1$^{B9}$ (PINK1$^{B9}$/+;; UAS-mt-Keima, mef-2-GAL4/+) and parkin1 RNAi ($w^{1118}$; UAS-parkin1 RNAi/+; UAS-mt-Keima, mef-2-GAL4/+) flies. (**B**) High (543/458) ratio area/total mitochondrial area was quantified as an index of mitophagy in 1-, 3- and 4-week-old flies (*n* = 3–6 flies per condition). In each fly, 7 random 2500 µm$^2$ fields were analyzed. *p=0.02 compared with 4-week-old *parkin1* RNAi and *PINK1$^{B9}$* flies. # p=0.03 compared with 3-week-old *parkin1* RNAi and *PINK1$^{B9}$* flies. (**C**) Overlay of live mt-Keima emission at 458 (green) and 543 nm (red) excitation in indirect flight muscle from 4-week-old Ctrl RNAi ($w^{1118}$; UAS-Ctrl RNAi/+; UAS-mt-Keima, mef-2-GAL4/+), dUSP15 RNAi ($w^{1118}$;; UAS-mt-Keima, mef-2-GAL4/UAS-CG8334 RNAi), dUSP30 RNAi ($w^{1118}$; UAS-CG3016 RNAi/+; UAS-mt-Keima, mef-2-GAL4/+), parkin1 +Ctrl RNAi ($w^{1118}$; UAS-parkin1 RNAi/UAS-Ctrl RNAi; UAS-mt-Keima, mef-2-GAL4/+), parkin1 +dUSP15 RNAi ($w^{1118}$; UAS-parkin1 RNAi/+; UAS-mt-Keima, mef-2-GAL4/UAS-CG8334 RNAi), parkin1 +dUSP30 RNAi ($w^{1118}$; UAS-parkin1 RNAi/UAS-CG3016 RNAi; UAS-mt-Keima, mef-2-GAL4/+), parkin2 +Ctrl RNAi ($w^{1118}$; UAS-parkin2 RNAi/UAS-Ctrl RNAi; UAS-mt-Keima, mef-2-GAL4/+), parkin2 +dUSP15 RNAi ($w^{1118}$; UAS-parkin2 RNAi/+; UAS-mt-Keima, mef-2-GAL4/UAS-CG8334 RNAi), parkin2 +dUSP30 RNAi ($w^{1118}$; UAS-parkin2 RNAi/UAS-CG3016 RNAi; UAS-mt-Keima, mef-2-GAL4/+), PINK1$^{B9}$B9Ctrl RNAi (PINK1$^{B9}$/+; UAS-Ctrl RNAi/

*Figure 3 continued on next page*

*Figure 3 continued*

+; *UAS-mt-Keima, mef-2-GAL4/+*), *PINK1^B9* + *dUSP15* RNAi (*PINK1^B9*/+;; *UAS-mt-Keima, mef-2-GAL4/UAS-CG8334 RNAi*), and *PINK1^B9* + *dUSP30* RNAi (*PINK1^B9*/+; *UAS-CG3016 RNAi/+; UAS-mt-Keima, mef-2-GAL4/+*) flies. (D) High (543/458) ratio area/total mitochondrial area was quantified as an index of mitophagy (n = 3–4 flies per condition). *p<0.001 compared with Ctrl RNAi flies. # p<0.03 compared with *parkin1* +*dUSP15* RNAi flies. £ p<0.03 compared with *PINK1^B9* + *dUSP15* RNAi flies. Scale bars, 10 µm.
DOI: https://doi.org/10.7554/eLife.35878.006
The following source data is available for figure 3:

**Source data 1.** Quantification of mitophagy (high [543/458] ratio area/total mitochondrial area) in indirect flight muscle of *w^1118*, control RNAi, *PINK1^B9* and *parkin1* RNAi flies at 1, 3 and 4 weeks.
DOI: https://doi.org/10.7554/eLife.35878.007
**Source data 2.** Quantification of mitophagy (high [543/458] ratio area/total mitochondrial area) in indirect flight muscle of 4-week-old control RNAi, *dUSP15* RNAi, *dUSP30* RNAi, *parkin1* + control RNAi, *parkin1* + *dUSP15* RNAi, *parkin1* + *dUSP30* RNAi, *parkin2* + control RNAi, *parkin2* + *dUSP15* RNAi, *parkin2* + *dUSP30* RNAi, *PINK1^B9* +control RNAi, *PINK1^B9* + *dUSP15* RNAi, and *PINK1^B9* + *dUSP30* RNAi flies.
DOI: https://doi.org/10.7554/eLife.35878.008

mitophagy in mice and may explain why PINK1 or parkin deficiency by itself does not cause a degenerative phenotype in this species.

In a recent *Drosophila* study Lee *et al.* were unable to detect mitophagy in flight muscle and found no evidence for a major role for PINK1 or parkin in mitophagy in other fly tissues (*Lee et al., 2018*). The discrepancy with our correlative mt-Keima and TEM imaging findings may be due to the fact that Lee et al. mostly relied on imaging with *mito*-QC, a probe that may be less sensitive for parkin-mediated forms of mitophagy, as discussed above. Also, Lee *et al.* restricted their analysis of flight muscle to 2-day-old or younger flies, and may thus have missed the PINK1/parkin-dependent rise in mitophagy that occurs at later ages.

Recent publications proposed a distinction between basal and induced mitophagy (*McWilliams et al., 2018*; *Lee et al., 2018*). According to this terminology, the physiological mitophagy observed in our study could be labeled as basal. However, this may be confusing, because the term 'basal' suggests a relatively stationary background phenomenon, whereas our data show that 'basal' mitophagy in the fly is strongly induced by normal aging.

We did not detect a significant reduction in mitophagy in 1-week-old *PINK1^B9* and *parkin* RNAi flies, at a time point when these flies already display mitochondrial abnormalities (*Greene et al., 2003*; *Cornelissen et al., 2014*). This suggests that the mitochondrial changes in 1-week-old *PINK1^B9* and *parkin* RNAi flies are caused by loss of aspects of PINK1 and parkin function that are unrelated to mitophagy. For example, parkin also regulates $Ca^{2+}$ transfer from ER to mitochondria (*Gautier et al., 2016*). PINK1 promotes mitochondrial complex I activity through phosphorylation of the complex I subunit NdufA10 (*Morais et al., 2014*) and is involved in crista junction remodeling via phosphorylation of dMIC60 (*Tsai et al., 2018*). Alternatively, the early mitochondrial abnormalities in PINK1- and parkin-deficient flies may be due to more subtle reductions in mitophagy that are not detectable with mt-Keima imaging.

Genetic manipulation in *Drosophila* is relatively straightforward. As illustrated by our *dUSP15* and *dUSP30* knockdown experiments, this novel mt-Keima fly model is a convenient tool to determine the impact of individual genes on PINK1/parkin-mediated mitophagy in vivo. This model will greatly facilitate the identification of targets for modulation of a pathway with growing relevance for neurodegenerative diseases.

# Materials and methods

**Key resources table**

| Reagent type (species) or resource | Designation | Source or reference | Identifiers | Additional information |
|---|---|---|---|---|
| Strain (*D. melanogaster*) | *parkin* KK RNAi | VDRC | 107919 RRID:FlyBase_FBst0476221 | Named *parkin2* RNAi in this paper |

*Continued on next page*

*Continued*

| Reagent type (species) or resource | Designation | Source or reference | Identifiers | Additional information |
|---|---|---|---|---|
| Strain (*D. melanogaster*) | *dUSP15* RNAi | VDRC | 18981 RRID:BDSC_61871 | |
| Strain (*D. melanogaster*) | *dUSP30* RNAi | NIG-Fly Stock Center | 3016 R-2 | |
| Strain (*D. melanogaster*) | control TRiP RNAi | BDSC | 31603 RRID:BDSC_31603 | |
| Strain (*D. melanogaster*) | *parkin* TRiP RNAi | BDSC | 37509 RRID:BDSC_37509 | Named *parkin1* RNAi in this paper |
| Strain (*D. melanogaster*) | *mef-2*-GAL4 | BDSC | 27390 RRID:BDSC_27390 | |
| Strain (*D. melanogaster*) | *TH*-GAL4 | BDSC | 8848 RRID:BDSC_8848 | |
| Strain (*D. melanogaster*) | $PINK1^{B9}$ | BDSC | 34749 RRID:BDSC_34749 | |
| Strain (*D. melanogaster*) | $Atg1^{K38A}$ | BDSC | 60736 RRID:BDSC_60736 | |
| Strain (*D. melanogaster*) | *VK20* | BDSC | 9738 RRID:BDSC_9738 | |
| Strain (*D. melanogaster*) | mt-Keima | This paper | | mt-Keima cDNA was cloned into the NotI and Xba1 sites of pUAS-attB and inserted in integration site VK20 after in-house injection |
| Antibody | ATP synthase β | Abcam | ab14730 RRID:AB_301438 | 1:500 |
| Recombinant DNA reagent | mt/mKeima/pIND(SP1) | *Katayama et al. (2011)* | | Gift from Dr. A. Miyawaki (RIKEN Brain Science Institute, Japan) |

## *Drosophila* genetics

All *Drosophila melanogaster* stocks and experimental crosses were kept on standard corn meal and molasses food at room temperature. The mt-Keima construct (mt/mKeima/pIND(SP1)) was a gift from Dr. A. Miyawaki (RIKEN Brain Science Institute, Japan) (*Katayama et al., 2011*). The mt-Keima cDNA was cloned into the NotI and Xba1 sites of pUAS-attB and inserted in integration site VK20 after in-house injection. *CG8334* transgenic UAS-RNAi (18981) and *parkin* KK UAS-RNAi (107919, named *parkin2* RNAi in this paper) lines were obtained from the Vienna Stock Center (VDRC) and *CG3016* transgenic UAS-RNAi line (3016 R-2) from NIG-Fly Stock Center. Parkin (37509, named *parkin1* RNAi in this paper) and control (luciferase, 31603) TRiP UAS-RNAi lines, *mef-2-GAL4*, *TH-GAL4*, $PINK1^{B9}$ and $Atg1^{K38A}$ were obtained from Bloomington stock center (Indiana, USA). To quantify *parkin*, *CG8334* and *CG3016* mRNA levels under control of *mef-2-GAL4*, RNA was isolated from adult thoraces and real-time RT-PCR was performed as previously described (*Cornelissen et al., 2014*) using primers 5'-CCAGCAATGTCACCATCAAAG-3' and 5'-GCGTGTCCACTCAGTCTG-3' for *parkin*, 5'-GGAGTGACGCATCTTGAG-3' and 5'-TTCTTTGGTATGGGTGGACTG-3' for *CG8334* and 5' TACGCCATAGCAATCTGGGG-3' and 5'- CTCGTGTATCTGCTGGCGTT-3' for *CG3016*. The data were normalized using *RP-49*, a ribosomal gene. Real-time PCR showed that *parkin* mRNA levels in *parkin1* RNAi flies were 49,7 ± 2,3% of control levels (*n* = 5). Parkin mRNA levels in *parkin2* RNAi flies were determined previously (*Cornelissen et al., 2014*).

## Live mt-Keima imaging

Flies were dissected in HL3 buffer. Indirect flight muscle fibers and complete brains were immobilized on a glass slide in low gelling temperature agarose and analyzed using a Leica TCS SP5 II confocal microscope equipped with a 63x objective lens (HC PL APO 63x/1.4 CS2), a multi-argon laser (458, 476, 488 nm) and a He/Ne laser (543 nm). Mt-Keima was imaged in two channels via two sequential excitations (458 nm, green; 543 nm, red) and using a 600 to 695 nm emission range. Images from random microscopic fields were captured and analyzed by an investigator blinded to genotype and age. For muscle analysis, at least 7 z-stacks with 0.2 µm slice thickness were taken per

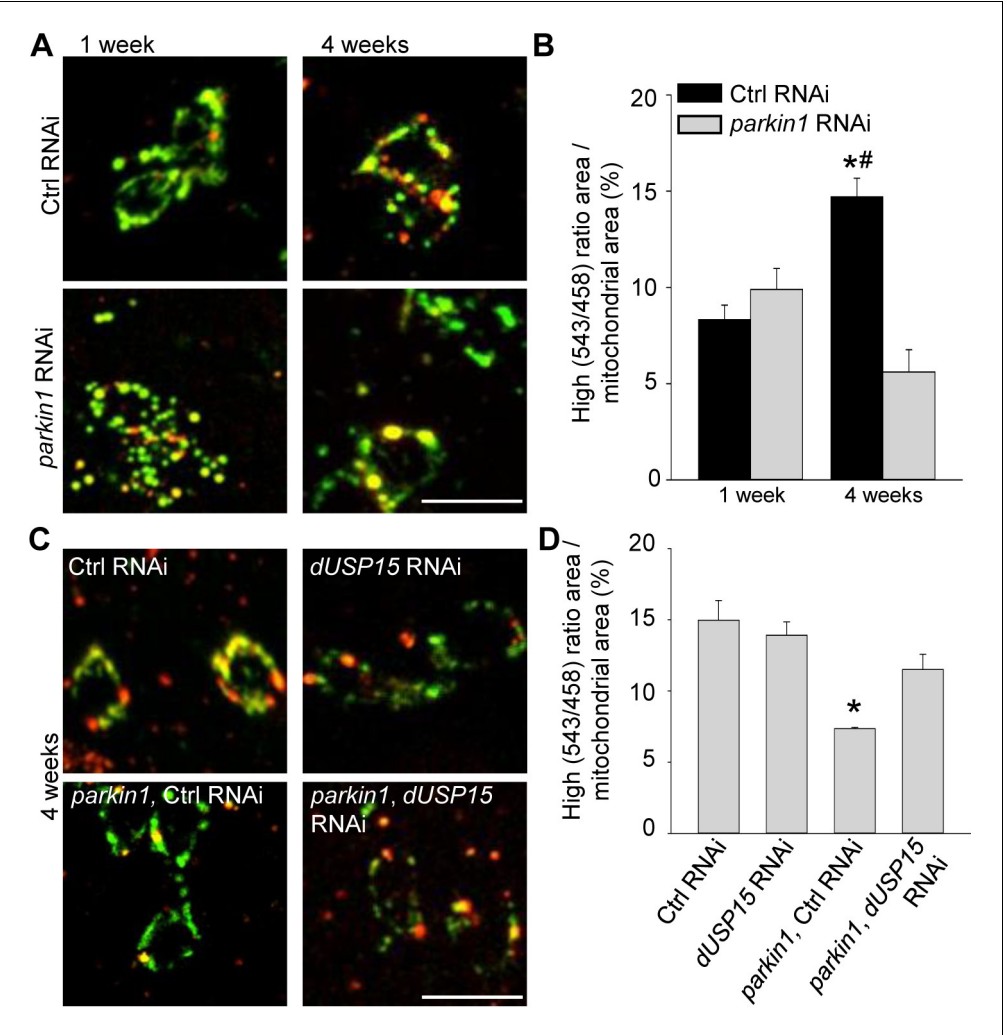

**Figure 4.** Parkin deficiency impairs mitophagy in dopaminergic neurons. (**A**) Overlay of live mt-Keima emission at 458 (green) and 543 nm (red) excitation in PPL1 dopaminergic neurons from 1- and 4-week-old control (Ctrl) RNAi ($w^{1118}$; *UAS*-Ctrl *RNAi/+*; *UAS-mt-Keima, TH-GAL4/+*) and *parkin1* RNAi ($w^{1118}$; *UAS-parkin1 RNAi/+*; *UAS-mt-Keima, TH-GAL4/+)* flies. (**B**) High (543/458) ratio area/total mitochondrial area was quantified as an index of mitophagy (*n* = 4 flies per condition). In each fly, 10 dopaminergic neurons were analyzed. *p=0.001 compared with 4-week-old *parkin* RNAi flies. # p=0.004 compared with 1-week-old Ctrl RNAi flies. (**C**) Overlay of live mt-Keima emission at 458 (green) and 543 nm (red) excitation in PPL1 dopaminergic neurons from 4-week-old Ctrl RNAi ($w^{1118}$; *UAS*-Ctrl *RNAi/+*, *UAS-mt-Keima, TH-GAL4/+*), *dUSP15* RNAi ($w^{1118}$;; *UAS-mt-Keima, TH-GAL4/UAS-CG8334 RNAi*), *parkin1* +Ctrl RNAi ($w^{1118}$; *UAS-parkin1 RNAi/UAS*-Ctrl RNAi; *UAS-mt-Keima, TH-GAL4/+*), and *parkin1* +*dUSP15* RNAi ($w^{1118}$; *UAS-parkin1 RNAi/+*; *UAS-mt-Keima, mef-2-GAL4/UAS-CG8334 RNAi*) flies. (**D**) High (543/458) ratio area/total mitochondrial area in 4-week-old flies was quantified as an index of mitophagy (*n* = 3–4 flies per condition). *p<0.05 compared with all other conditions. Scale bars, 10 μm.
DOI: https://doi.org/10.7554/eLife.35878.009

The following source data is available for figure 4:

**Source data 1.** Quantification of mitophagy (high [543/458] ratio area/total mitochondrial area) in PPL1 dopaminergic neurons from control RNAi and *parkin1* RNAi flies at 1 and 4 weeks.
DOI: https://doi.org/10.7554/eLife.35878.010
**Source data 2.** Quantification of mitophagy (high [543/458] ratio area/total mitochondrial area) in PPL1 dopaminergic neurons from 4-week-old control RNAi, *dUSP15* RNAi, *parkin1* +control RNAi, and *parkin1* +*dUSP15* RNAi flies.
DOI: https://doi.org/10.7554/eLife.35878.011

fly. For dopaminergic neuron analysis, at least 10 PPL1 neurons were analyzed per fly. Ratio (543/458) images were created using the Ratio Plus plugin in ImageJ. High (543/458) ratio areas were

segmented and quantified with the Analyze Particles plugin in ImageJ. The total mitochondrial area was quantified with the Analyze Particles plugin by calculating the area of the total emission at 543 nm excitation. The parameter (high [543/458] ratio area/total mitochondrial area) was used as an index of mitophagy, as described (*Katayama et al., 2011*). LysoTracker (DND-26, Thermo Fisher) was imaged using a 488 nm excitation and a 495–550 nm emission filter.

### Immunohistochemistry

Indirect flight muscle was dissected in HL3, live imaged for mt-Keima and fixed in 4% paraformaldehyde for 1 hr. Samples were washed three times for 10 min in PBS and placed in blocking solution (5% normal donkey serum in PBS, 0.1% Triton X-100) for 30 min, after which the samples were incubated overnight at 4°C with antibody against ATP synthase β (1:500; ab14730, Abcam) in PBS, 2% normal donkey serum, 0.1% Triton X-100. After washing, samples were incubated for 2 hr with Alexa 488-conjugated secondary antibody in PBS, 2% normal donkey serum, 0.1% Triton X-100. Samples were washed and mounted on a glass slide in Vectashield mounting medium (Vector Laboratories). Alexa 488 fluorescence was imaged using a 488 nm excitation and a 495–550 nm emission filter.

### CLEM

CLEM studies using near-infrared branding (NIRB) were performed as described previously (*Bishop et al., 2011*; *Urwyler et al., 2015*). Tissues were first imaged live in low gelling temperature agarose (3% in 0.1 M phosphate buffer, pH 7.4 [PB]), and fixed overnight in 0.5% glutaraldehyde, 4% PFA in 0.1 M PB, pH 7.4 at 4°C. Samples were rinsed in 0.1 M PB after which fixed samples were imaged using the Leica SP5 confocal microscope to check whether fixing affected overall tissue morphology. The bleaching function of the ZEN2010 software (Zeiss) was used to perform NIRB on a LSM 780 inverted confocal microscope. Branding marks were introduced to the tissue with a Mai Tai DeepSee two-photon laser (Spectra-Physics) at 800 nm and 40% maximal power output. Z-stacks of the region of interest were acquired before and after NIRB. Processed samples were post-fixed in 2% glutaraldehyde, 4% paraformaldehyde and 0.2% picric acid in 0.1 M PB at 4°C overnight or until further processing. Next, samples were osmicated for 1 hr in 2% OsO4, 1.5% potassium ferrocyanide in PB and subsequently stained for 30 min in 0.2% tannic acid, followed by overnight incubation in 0.5% uranyl acetate in 25% methanol. Next day, samples were stained *en bloc* with lead aspartate and dehydrated in an ascending series of ethanol solutions followed by flat embedding in Agar 100 (Laborimpex; Agar Scientific). Flat-embedded sections were mounted on aluminum pin stubs (Gatan) with conductive epoxy (Circuit Works). A Zeiss Sigma Variable pressure SBF-SEM with 3View technology (Gatan) was used to approach the region of interest. Imaging was done at 1.3 kV with a pixel size of 20 nm and sections of 200 nm. Regions of interest could be located based on the branding marks and muscle fiber morphology. When the region of interest was reached, 70 nm thick, serial ultrathin sections were cut using a Reichardt Ultracut E ultramicrotome. All sections were collected as ribbons of 4–5 sections on triple slot grids (Ted Pella). Images were taken on a JEOL JEM-1400 Transmission Electron Microscope operated at 80 kV.

### Statistics

Values and error bars represent mean ± SEM, and *n* refers to the number of biological replicates. Significance of differences between conditions was analyzed with one-way ANOVA and post-hoc Holm-Sidak test (SigmaStat 3.5, Systat). No outliers were excluded.

## Acknowledgements

TC is a Postdoctoral Fellow and WV a Senior Clinical Investigator of the Research Foundation Flanders (FWO). This work was supported by FWO (Research Grant 1500817N) and KU Leuven ('Opening the Future' Campaign). We are grateful to A Miyawaki for providing the mt-Keima construct. We thank A Geens for injecting the mt-Keima-pUAS-attB construct.

## Additional information

### Competing interests
Patrik Verstreken: Reviewing editor, *eLife*. The other authors declare that no competing interests exist.

### Funding

| Funder | Grant reference number | Author |
| --- | --- | --- |
| Research Foundation - Flanders | 1500817N | Tom Cornelissen |
| KU Leuven | | Wim Vandenberghe |

The funders had no role in study design, data collection and interpretation, or the decision to submit the work for publication.

### Author contributions
Tom Cornelissen, Conceptualization, Data curation, Formal analysis, Funding acquisition, Investigation, Methodology, Writing—review and editing; Sven Vilain, Katlijn Vints, Natalia Gounko, Methodology, Writing—review and editing; Patrik Verstreken, Conceptualization, Supervision, Methodology, Writing—review and editing; Wim Vandenberghe, Conceptualization, Formal analysis, Supervision, Funding acquisition, Methodology, Writing—original draft, Writing—review and editing

### Author ORCIDs
Wim Vandenberghe (iD) http://orcid.org/0000-0002-9758-5062

### Decision letter and Author response
Decision letter https://doi.org/10.7554/eLife.35878.014
Author response https://doi.org/10.7554/eLife.35878.015

## Additional files

### Supplementary files
• Transparent reporting form
DOI: https://doi.org/10.7554/eLife.35878.012

### Data availability
All data generated or analysed during this study are included in the manuscript and supporting files. Source data files have been provided for Figures 1, 3 and 4.

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
