## [Decision Letter]

Thank you for submitting your article "Deficiency of parkin and PINK1 impairs age-dependent mitophagy in *Drosophila*" for consideration by *eLife*. Your article has been favorably evaluated by Ivan Dikic (Senior Editor) and three reviewers, one of whom is a member of our Board of Reviewing Editors.

The reviewers have discussed the reviews with one another and the Reviewing Editor has drafted this decision to help you prepare a revised submission.

Summary:

The manuscript of Cornelissen et al. is an interesting, well-written, and well-developed study that addresses an important and controversial question: does the well-established pathway by which PINK1 and parkin can trigger mitophagy actually matter to neurons in vivo. Despite all the mechanistic analysis of this pathway in vitro, evidence of its importance in the normal context in vivo has been scarce and depended on the presence of additional stressors such as toxins or mutator strains. Negative results have been reported in the mouse and, very recently, in *Drosophila* as well. The present manuscript offers very persuasive data to say that it does function in vivo but only in aged animals, in this case *Drosophila*. Among the strong points of the manuscript are the care with which they have justified the use of their reporter, mito-Keima as a marker of mitochondria and, when the fluorescence ratio has shifted, as a marker of mitochondrial proteins or mitochondria-like structures in electron-dense lysosomes, including by CLEM. Other strong points include the use of a dominant negative ATG1 to show that this signal is autophagy driven, the use of DUB RNAi to confirm that the mitophagy is happening by the predicted mechanism of Parkin, clear statistical analysis and the inclusion of the actual P values, and a thoughtful discussion of why the other recent studies failed to see the change in mitophagy with PINK or Parkin mutants.

Essential revisions:

Two areas of concern were noted that require additional data.

1) Figure 1C is problematic because of the spectral overlap of the fluorophores. Rather than the use of a second GFP reporter, an antibody to a mitochondrial protein should be used and then visualized with a fluorescent secondary antibody in a frequency range far from those of Keima.

2) The section on the deubiquitinases needs strengthening. This needs the inclusion of a proper control UAS-construct to exclude the possibility that the UAS-USP15 and USP30 genes are not merely titrating GAL4 away from the UAS-Parkin RNAi. Inclusion of a Parkin hypomorphic mutation, rather than only the RNAi, and of a PINK1 allele in Figure 3C, D would also strengthen the analysis.

Additional concerns can be addressed in the text.

1) A more detailed description of the image processing used throughout.

2) The authors are commended for discussing the discrepancies between their study and that of Lee et al., but are not always accurate in describing that work. Among other things, they should clarify the distinction between basal and induced mitophagy.

3) The authors should address the unlikely but formal possibility that increased Keima signal results from slower degradation of Keima in the mutants rather than increased mitophagy.

---

## [Author Response]

Essential revisions:Two areas of concern were noted that require additional data.1) Figure 1C is problematic because of the spectral overlap of the fluorophores. Rather than the use of a second GFP reporter, an antibody to a mitochondrial protein should be used and then visualized with a fluorescent secondary antibody in a frequency range far from those of Keima.

As requested by the reviewer, we have removed the mito-GFP data from the manuscript. We have now performed immunohistochemistry on indirect flight muscle using a primary antibody against the mitochondrial protein ATP synthase b and an Alexa 488-conjugated secondary antibody. In their original paper on Keima and mt-Keima imaging (Katayama et al., 2011) Katayama et al. demonstrate that the Keima and mt-Keima probes can be imaged simultaneously with the Alexa 488 fluorophore within the same cell without serious concerns for cross-excitation, cross-detection and resonance transfer (cfr. Figures S3, S4 and S5 in their paper).

Our experiments showed that the mt-Keima signal in flight muscle extensively colocalized with ATP synthase b, confirming that mt-Keima was properly targeted to mitochondria. We show these data in the new panel Figure 1C.

2) The section on the deubiquitinases needs strengthening. This needs the inclusion of a proper control UAS-construct to exclude the possibility that the UAS-USP15 and USP30 genes are not merely titrating GAL4 away from the UAS-Parkin RNAi. Inclusion of a Parkin hypomorphic mutation, rather than only the RNAi, and of a PINK1 allele in Figure 3C, D would also strengthen the analysis.

- We have now included proper control UAS constructs in Figure 3C-D and Figure 4C-D. The data show that there was no effect of “GAL4 dilution” and thus the conclusion of the experiment did not change.

- In *Drosophila* the parkin gene is located on the third chromosome. Unfortunately, the *dUSP15* RNAi, *mef-2-GAL4* and mt-Keima transgenes are also located on the third chromosome. We were not able to recombine these 3 insertions with a hypomorphic parkin mutation (all 4 are on the third chromosome).

We nonetheless provide further support by instead including data obtained with an additional *parkin* RNAi line (*parkin* KK RNAi). In the new Figure 3C-D we show that knockdown of *dUSP15* and *dUSP30* rescues the mitophagy defect of 4-week-old flies that express *parkin* KK RNAi (similar to the rescue of flies expressing *parkin* TRiP RNAi).

As requested, we also included the *PINK1^B9^*flies in the section on the deubiquitinases. We found that knockdown of *dUSP15* partially rescued the mitophagy defect of 4-week-old *PINK1^B9^*flies, while knockdown of *dUSP30* had no effect. We have added these findings to Figure 3C-D.

Additional concerns can be addressed in the text.1) A more detailed description of the image processing used throughout.

We have added more details on image processing to the Materials and methods (section ‘Live mt-Keima imaging’).

2) The authors are commended for discussing the discrepancies between their study and that of Lee et al., but are not always accurate in describing that work. Among other things, they should clarify the distinction between basal and induced mitophagy.

We have revised our description of the study by Lee et al. so that the description is fully accurate.

Recent publications by Lee et al. (2018) and by McWilliams et al. (2018) indeed proposed a distinction between basal and induced mitophagy. According to this terminology, the physiological mitophagy observed in our study could probably be labeled as basal. However, this terminology may be confusing, because the term ‘basal’ suggests a relatively stationary background phenomenon, whereas our data show that ‘basal’ mitophagy in the fly is strongly induced by normal aging.

We have added these comments to the Discussion of the revised manuscript.

3) The authors should address the unlikely but formal possibility that increased Keima signal results from slower degradation of Keima in the mutants rather than increased mitophagy.

In their original paper on Keima and mt-Keima imaging (Katayama et al., 2011) Katayama et al. convincingly demonstrate that lysosomal proteases are to degrade Keima and mt-Keima (Figure S2 in their paper). The known resistance of mt-Keima to lysosomal degradation makes it extremely unlikely that the increased lysosomal mt-Keima signal in the parkin- and PINK1-deficient flies results from slower degradation of mt-Keima in the mutant lysosomes.